# VeriReason: Reinforcement Learning with Testbench Feedback for Reasoning-Enhanced Verilog Generation

## Abstract

Automating Register Transfer Level (RTL) code generation using Large Language Models (LLMs) offers substantial promise for streamlining digital circuit design and reducing human effort. However, current LLM-based approaches for RTL code generation face significant challenges. Methods such as supervised fine-tuning (SFT), in-context learning, and chain-of-thought (CoT) struggle with several critical limitations in the RTL domain: the scarcity of high-quality training data, poor alignment between natural language specifications and generated code, lack of built-in verification mechanisms, and difficulty balancing between model generalization and domain specialization. Inspired by groundbreaking research such as DeepSeek-R1, which combines reinforcement learning with reasoning capabilities, we introduce `VeriReason`, a comprehensive framework that integrates supervised fine-tuning with Group Relative Policy Optimization (GRPO) reinforcement learning specifically tailored for RTL code generation. Using our curated high-quality training examples alongside a feedback-driven reward model, `VeriReason` combines testbench evaluations with structural heuristics to improve specification-code alignment and eliminate hallucinations. Iterative GRPO embeds intrinsic self-checking and reasoning capabilities, enabling the model to autonomously detect and correct functional errors. On the VerilogEval Benchmark, `VeriReason` delivers significant improvements: achieving 83.1% functional correctness on the VerilogEval Machine benchmark, substantially outperforming both comparable-sized models and much larger commercial systems like GPT-4 Turbo. Additionally, our approach demonstrates up to a 2.8× increase in first-attempt functional correctness compared to baseline methods and exhibits robust generalization to unseen designs. To our knowledge, `VeriReason` represents the first system to successfully integrate explicit reasoning capabilities with reinforcement learning for Verilog generation, establishing a new state-of-the-art for automated RTL synthesis. The code is available at: https://anonymous.4open.science/r/VeriReason-E625.

## 1 Introduction

Register Transfer Level (RTL) code generation is a critical yet labor-intensive task in digital circuit design, directly impacting the efficiency, performance, and power consumption of hardware systems. Traditionally, hardware engineers manually craft RTL code using hardware description languages (HDLs) such as Verilog, which differs significantly from general-purpose programming languages due to its concurrent and structural nature. Recent advancements in large language models (LLMs) offer promising opportunities to automate RTL code generation, substantially reducing the manual effort and domain expertise required. Leveraging LLMs for RTL generation can accelerate design cycles, minimize human-induced errors, and allow engineers to focus on high-level architectural decisions rather than intricate coding details.

Despite these advantages, LLM-based RTL synthesis encounters three core challenges. First, **data scarcity**: high-quality Verilog examples—and especially paired testbenches or reasoning annotations—are rare, limiting both pretraining and supervised fine-tuning (SFT) and hampering generalization. Second, **weak natural language–code alignment**: LLMs often produce syntactically valid but functionally incorrect Verilog, misinterpreting user specifications and hallucinating invalid structural

heuristics (*e.g.*, port matching, net connectivity). Third, **low first-attempt accuracy without self-checking**: current models lack intrinsic mechanisms to detect or correct their own errors, relying instead on external testbench or syntax feedback for iterative refinement. Lastly, **lack of complex logical capability:** Traditional LLMs struggle to handle the intricate interdependencies between components in hardware design, often failing to maintain consistency across module interfaces, state machines, and timing constraints. Without systematic reasoning about component relationships, models produce circuits with logical inconsistencies or incomplete implementations that meet superficial requirements but fail under comprehensive verification.

Recent advances in reasoning and reinforcement learning (RL) have introduced promising approaches to overcome these challenges. Reasoning-augmented models, such as those leveraging chain-of-thought prompting or iterative refinement, have demonstrated the ability to follow multi-step logical patterns, making them particularly suitable for hardware description languages like Verilog that require strict structural correctness and functional dependencies. These reasoning mechanisms help LLMs better understand circuit intent and adhere to design constraints, and can better ensure the alignment between natural language and result. Methods such as Group Relative Policy Optimization (GRPO)Shao et al. (2024) combine the strengths of SFT with reward-driven RL, enabling models to learn effectively even with minimal data and explicit feedback. By employing RL-based strategies, LLMs are trained not merely on predicting the next token but on achieving specific, meaningful outcomes, thus improving their logical reasoning, alignment, and self-checking capabilities.

**Our Proposed Framework.** To address the challenges in RTL generation with LLMs, we propose a novel framework, `VeriReason`, combining supervised fine-tuning (SFT) and GRPO reinforcement learning, specifically tailored for Verilog RTL generation with a specially designed dataset featuring reasoning steps and testbenches. Our approach systematically tackles four critical limitations that hinder existing LLM-based hardware design methods: data scarcity in domain-specific code, natural language-code alignment issues, lack of self-checking behavior, and insufficient complex logical capabilities. Each of these challenges requires specialized techniques that we incorporate into the `VeriReason` framework, as detailed in the following sections.

*Data Scarcity in Domain-Specific Code:* We introduce a reasoning-distillation and testbench-generation pipeline to augment existing prompt–code pairs with high-quality testbenches and human-style reasoning steps, producing a high-quality dataset. Furthermore, we demonstrate that even with as few as 20 annotated examples from the `VeriReason` dataset, GRPO yields substantial performance gains, dramatically lowering the bar for required training data.

*Natural Language-Code Alignment:* `VeriReason` employ a reward model that evaluates generated Verilog code against specifications using feedback from structural heuristics. Through GRPO optimization, the model learns to internalize structural constraints, effectively reducing hallucinations and ensuring structural correctness by penalizing invalid constructs across both interface definitions and internal hierarchy of the circuit design.

*Lack of Self-Checking Behavior:* Our reinforcement learning framework inherently encourages the model to develop self-checking capabilities by iteratively refining outputs based on testbench feedback-driven rewards. Over training iterations, the model learns to anticipate and rectify errors internally, significantly enhancing first-attempt functional correctness.

*Lack of complex logical capabilities:* `VeriReason` incorporates explicit reasoning steps throughout the design process, requiring the model to articulate its design decisions and verify logical consistency before implementation. By decomposing complex circuit specifications into manageable conceptual components and reasoning about their interactions, the model develops more coherent and complete implementations that maintain logical integrity across the entire design.

Our key contributions are summarized as follows:

- We design a novel framework `VeriReason`, which integrates supervised fine-tuning with GRPO-based reinforcement learning and reasoning-augmented design processes for Verilog RTL code generation.

- Our approach addresses critical shortcomings in RTL generation through a reasoning-distillation pipeline for data scarcity, reward-driven structural evaluation for NL-code alignment, testbench feedback mechanisms for self-checking behavior, and explicit reasoning steps for handling complex logical dependencies. These techniques can also generalize be-

yond RTL to other structured generation domains requiring multi-level correctness validation and logical reasoning.

- We create a high quality dataset with reasoning and testbench that would be open-sourced to the benefit community.

- The framework achieves state-of-the-art performance in RTL generation tasks, demonstrating substantial improvements in first-attempt functional correctness, structural validity, and generalization capabilities with minimal training data. It delivers up to a 2.8× increase in first-attempt functional correctness compared to baseline models while outperforming existing state-of-the-art methods across multiple benchmarks. This improvement is particularly notable in smaller parameter models. Remarkably, the framework achieves 83.1% pass@5 on VerilogEval-Machine, even surpassing much larger models including GPT-4 Turbo.

## 2 BACKGROUND

Recent years have seen a surge of interest in applying large language models (LLMs) to hardware design, particularly for generating Register-Transfer Level (RTL) code in Verilog.Liu et al. (2023a; 2024b); Chang et al. (2023); Blocklove et al. (2023); Wang et al. (2025b); Thakur et al. (2024); Zhong et al. (2023); Lu et al. (2024); Liu et al. (2024a); Tsai et al. (2024); Pearce et al. (2020); Fu et al. (2023). Previous reseach have shown significant potential for LLM-based RTL generation in automating parts of the hardware design process using finetuning, or using differnt prompting techniques. However, it has also revealed several key challenges of produce correct and efficient hardware designs reliably.

**Challenges in LLM-based RTL generation tasks** Researchers investigated fine-tuning LLMs using domain-specific data and techniques to improve their performance Liu et al. (2024b); Pei et al. (2024). For example, Liu et.al introduced ChipNeMo Liu et al. (2023a), which fine-tunes a general-purpose LLM on internal NVIDIA datasets for various chip design tasks. Similarly, Thankur et.al developed VeriGen Thakur et al. (2024) to improve Verilog generation capabilities. Subsequent works, such as RTLCoder Liu et al. (2024b) is trained based on automatically generated datasets. BetterV Pei et al. (2024), finetunes LLM by converting Verilog code to the C language. While effective, these methods face challenges of scalability and generalizability due to their high demand for high quality instruction-code pairs. The inherent limitation of lack of human-written RTL code and the low quality of generated code make it hard to make further improvement.

Moreover, LLM generated Verilog code often face the issue of hallucination. Prompt-based methods Blocklove et al. (2023); Chang et al. (2023) rely heavily on the quality and clarity of the input prompts, facing difficulties in consistently aligning complex, multi-step circuit specifications with the generated code. These methods often suffer from hallucinations or syntactically correct yet functionally incorrect outputs due to inadequate contextual understanding. Moreover, they inherently lack iterative refinement capabilities, making them incapable of progressively improving RTL code quality. Chain of Thought (CoT) methods Yang et al. (2025), which encourage models to generate step-by-step reasoning sequences, typically excel in structured reasoning tasks but face challenges in RTL contexts due to the strict requirement for functional correctness and structural precision. Although CoT enhances reasoning, its effectiveness heavily depends on the clarity and correctness of intermediate reasoning steps, which can still suffer from errors in the absence of explicit correctness feedback mechanisms. Furthermore, the CoT has made the process of generation very ineffective due to long inference time.

### 2.1 REINFORCEMENT LEARNING FOR LLM REASONING

Recent research has demonstrated the potential of reinforcement learning (RL) techniques to significantly enhance the reasoning capabilities of large language models (LLMs) Ouyang et al. (2022); Shao et al. (2024); Guo et al. (2025). By providing explicit rewards for logical correctness and step-wise reasoning, RL enables models to autonomously discover effective problem-solving strategies, often mirroring structured human reasoning Wei et al. (2022); Xie et al. (2025). Applications span mathematical problem solving (where RL fine-tuning on step-by-step correctness or final answer accuracy yields substantial improvements Shao et al. (2024); Guo et al. (2025)) and code generation,

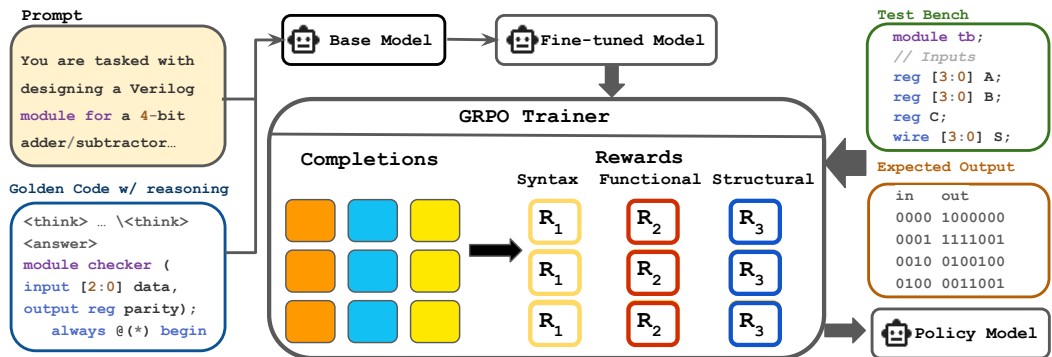

Figure 1: Workflow of `VeriReason`. The framework combines supervised fine-tuning with GRPO reinforcement learning. A base model is fine-tuned and then improved through the GRPO trainer, which leverages multiple reward signals (syntax, functional, and structural correctness) derived from testbench execution and code analysis. The model incorporates explicit reasoning (<think> blocks) to break down complex hardware design tasks.

where preference optimization and RL from feedback have led to greater code validity and efficiency Chen et al. (2021).

Most successful approaches build upon policy gradient algorithms such as Proximal Policy Optimization (PPO) Schulman et al. (2017) or, more recently, GRPO Shao et al. (2024); Lambert et al. (2024). GRPO, in particular, compares groups of generated responses rather than evaluating them in isolation, enabling the model to build a deeper understanding of what constitutes high-quality reasoning through relative comparisons. The effectiveness of these frameworks depends on carefully designed reward functions that accurately reflect the target domain. For hardware description tasks like Verilog generation, structural similarity as provides a clear, unambiguous reward signal Wang et al. (2025a), encouraging models to internalize domain-specific constraints and develop robust reasoning capabilities that translate natural language specifications into golden-code similarity.

## 3 METHODOLOGY

We present `VeriReason`, a comprehensive framework that combines supervised fine-tuning with reinforcement learning specifically tailored for Verilog RTL generation. Our approach addresses four critical challenges in automated hardware design: (1) data scarcity of high-quality RTL examples, (2) weak alignment between natural language specifications and generated code, (3) lack of self-checking mechanisms in current models, and (4) insufficient complex logical reasoning capabilities for hardware design.

As shown in Figure 1, `VeriReason` employs a multi-stage approach to generate high-quality Verilog code. First, a base model is fine-tuned on a curated dataset consisting of high-quality prompt-code pairs enhanced with explicit reasoning steps and testbenches. This fine-tuned model produces initial code implementations that are then evaluated through our reward system, which combines three key components: syntax correctness, functional validation via testbench execution, and structural analysis. The GRPO trainer leverages these rewards to iteratively improve the model's ability to generate correct code while maintaining alignment with the original specification. Through this process, the model learns to incorporate reasoning steps (shown as <think> blocks) that decompose complex hardware design problems into manageable components, resulting in a policy model capable of generating functionally correct and structurally sound Verilog implementations on the first attempt.

### 3.1 REINFORCEMENT LEARNING FRAMEWORK

Our approach adapts GRPO specifically for RTL code generation. Unlike traditional RL methods, our framework incorporates domain-specific constraints and verification mechanisms directly into the learning process, providing immediate feedback on functional correctness, syntax, and specifi-

cation adherence. This targeted optimization enables the model to efficiently learn correct Verilog implementation patterns while minimizing hallucinations and specification misalignments.

### 3.1.1 GROUP RELATIVE POLICY OPTIMIZATION (GRPO)

We adopt GRPO as our core reinforcement learning algorithm due to its efficiency and demonstrated effectiveness in tasks requiring complex reasoning. GRPO provides several advantages over traditional reinforcement learning methods like Proximal Policy Optimization (PPO), including lower memory requirements and more stable training dynamics.

In GRPO, the language model serves as the policy network, taking a natural language specification $q$ as input and producing a sequence of tokens representing Verilog code as actions. The policy distribution factors across tokens: $\pi_\theta(a|q) = \prod_{t=1}^{N} \pi_\theta(a_t|q, a_{<t})$, where $\pi_\theta$ represents the policy parameterized by $\theta$, $a$ is the complete sequence of tokens (the Verilog code), and $a_t$ is the token at position $t$.

Unlike PPO, which requires a separate value function, GRPO estimates advantages using group-based sampling. For each natural language specification $q$, we generate a group of $G$ candidate Verilog implementations $\{o_1, o_2, \ldots, o_G\}$ from the current policy and compute rewards for each. The GRPO objective function is defined as:

$$\mathcal{L}_{\text{GRPO}}(\theta) = \mathbb{E}_{q \sim \mathcal{D}, \{o_i\}_{i=1}^{G} \sim \pi_{\theta_{\text{old}}}(\cdot|q)} \left[ \frac{1}{G} \sum_{i=1}^{G} \min\left(r_i \cdot \rho_i, \text{clip}(\rho_i, 1 - \epsilon, 1 + \epsilon) \cdot r_i\right) \right] - \beta \cdot D_{\text{KL}}(\pi_\theta(\cdot|q) \| \pi_{\text{ref}}(\cdot|q)) \tag{1}$$

where: where $\rho_i = \frac{\pi_\theta(o_i|q)}{\pi_{\theta_{\text{old}}}(o_i|q)}$ is the importance sampling ratio, $r_i$ is the normalized reward for candidate $o_i$, $\epsilon$ is a hyperparameter controlling the clipping range, $\beta$ is a coefficient balancing the KL divergence penalty, $\pi_{\text{ref}}$ is a reference policy (typically the supervised fine-tuned model), and $D_{\text{KL}}$ is the Kullback-Leibler divergence.

The advantage estimation in GRPO is simplified by normalizing rewards within each group, where $r_i = \frac{R(o_i) - \mu_R}{\sigma_R + \delta}$, with $R(o_i)$ being the raw reward for output $o_i$, $\mu_R$ and $\sigma_R$ are the mean and standard deviation of rewards within the group, and $\delta$ is a small constant for numerical stability.

This group-based normalization provides several benefits: it eliminates the need for a separate value network, reduces variance in advantage estimation, and naturally compares alternative implementations of the same specification, which aligns well with the goal of generating functionally correct Verilog code. See Appendix B for theoretical analysis of GRPO's effectiveness with multi-level rewards compared to binary reward methods.

### 3.2 REWARD MODEL

Our reward function combines both structural correctness and functional validation to provide comprehensive feedback during training. The reward $R$ for a generated Verilog implementation is computed as:

$$R(o) = \begin{cases} 2.0, & \text{if functionally correct} \\ 0.1 + 1.0 \cdot \text{AST}_{\text{score}}(o), & \text{if syntactically correct} \\ 0, & \text{otherwise} \end{cases} \tag{2}$$

where: Functional correctness is determined by running the generated code through testbenches and comparing outputs with the expected behavior. Syntactic correctness is verified by successful parsing of the Verilog code. The $\text{AST}_{\text{score}}(o)$ measures structural similarity between the generated code's Abstract Syntax Tree (AST) and reference implementations, with values ranging from 0 to 1. The reward values establish a clear hierarchy: 0.1 for syntax-only correctness prevents gradient vanishing, the AST score (0.1-1.1) provides fine-grained structural feedback, and 2.0 for functional correctness creates a strong convergence signal approximately 2× the maximum partial reward. This ratio empirically balanced exploration with exploitation during GRPO training.

The AST score provides a fine-grained measure of structural correctness even when the code is not functionally perfect. `VeriReason` employs a hierarchical AST comparison algorithm specifically tailored for Verilog code structures, where $\text{ASTscore}(o) = \sum_{c \in C} w_c \cdot (0.6 \cdot \text{sim}_c + 0.5 \cdot \text{cov}_c - 0.3 \cdot \text{red}_c)$ calculates weighted structural similarity across categories $C = \{\text{module, port, always, ...}\}$ with respective importance weights $w_c$. For each category $c$, we compute sequence similarity $\text{sim}_c$ using Levenshtein distance, coverage $\text{cov}_c = |G_c \cap D_c|/|G_c|$ between generated elements $D_c$ and golden elements $G_c$, and redundancy $\text{red}_c = |D_c - G_c|/|D_c|$ to penalize hallucinated structures. This domain-specific structural analysis enables our model to maintain correct interface definitions, signal declarations, and control logic while internalizing hardware design patterns during GRPO optimization. This domain-specific structural analysis enables our model to maintain correct interface definitions, signal declarations, and control logic while internalizing hardware design patterns during GRPO optimization. The detailed AST extraction and scoring algorithms are provided in Appendix C. For functional verification, we use testbenches to evaluate the generated Verilog against reference implementations. A generated design is considered functionally correct only when it passes all test cases in the testbench, providing identical output signals to those of the golden reference for all test vectors.

### 3.3 Data Preprocessing

We address the critical issue of data scarcity in RTL generation, and the challenge of low dataset quality through a data augmentation pipelines, and a data filtration pipeline.

### 3.3.1 Data Filtration

We implement a two-stage adaptive filtration process to optimize the dataset for GRPO training effectiveness. For each sample $s$ in our initial dataset $D$, we generate a set of $k = 8$ candidate implementations $\{o_1, o_2, \ldots, o_k\}$ using our base model and compute their corresponding rewards $\{r_1, r_2, \ldots, r_k\}$ based on our reward function $R$ defined in Equation 2. The filtration process is formalized as follows: $D_{\text{filtered}} = \{s \in D \mid \mu_r(s) \in [\alpha_{\min}, \alpha_{\max}] \text{ and } \sigma_r(s) > \beta\}$, where: $\mu_r(s) = \frac{1}{k} \sum_{i=1}^{k} r_i$ is the mean reward for sample $s$, $\sigma_r(s) = \sqrt{\frac{1}{k} \sum_{i=1}^{k} (r_i - \mu_r(s))^2}$ is the standard deviation of rewards, $\alpha_{\min} = 0.3$ is the minimum acceptable mean reward, $\alpha_{\max} = 1.8$ is the maximum acceptable mean reward, $\beta = 0.1$ is the minimum acceptable reward variance. This filtration strategy excludes samples that are either too difficult (consistently low rewards) or too trivial (consistently high rewards), retaining only those samples that provide meaningful learning signals for the GRPO algorithm. Specifically, we filter out data where all rewards are zero or the average reward is below $\alpha_{\min}$, as well as samples where all generations achieve near-perfect scores.

Additionally, we compute a difficulty score $\delta(s)$ for each remaining sample:

$$\delta(s) = 1 - \frac{\mu_r(s) - \alpha_{\min}}{\alpha_{\max} - \alpha_{\min}} \tag{3}$$

Samples are then categorized into "simple" and "hard" portions based on this difficulty score, with samples where $\delta(s) > 0.5$ classified as "hard" and the remainder as "simple." This stratification enables targeted training strategies that progressively build model competence. The final dataset comprises 1149 samples in the hard level and 743 samples in the easy level, example in Appendix E.

### 3.3.2 Reasoning Generation with Optimization

To enhance the model's reasoning capabilities, we augment the training data with explicit reasoning steps that decompose complex hardware design problems into manageable components. Our reasoning generation pipeline extracts natural language specifications and corresponding Verilog implementations from the original dataset and uses chain-of-thought prompting with domain-specific guidance to generate detailed reasoning steps that explain the design choices, module interfaces, and implementation details. We apply an optimization process where the model is encouraged to critique its own reasoning and suggest improvements. When the model identifies a better alternative solution, we generate an improved implementation and incorporate it into the dataset. This self-improvement mechanism enables the model to iteratively refine both its reasoning process and the quality of

generated code. The resulting dataset contains not only input-output pairs but also explicit reasoning traces that guide the model to develop stronger internal reasoning capabilities.

### 3.3.3 TESTBENCH GENERATION

To provide reliable functional correctness signals during training, we develop an automated testbench generation pipeline that analyzes specifications to identify signal characteristics, boundary conditions, and expected behaviors. The pipeline generates comprehensive testbenches with at least 100 test vectors per specification, combining directed testing for explicit requirements with constrained random testing for broader coverage. These testbenches serve dual purposes: computing rewards during GRPO training and validating final outputs, ensuring the model produces functionally correct Verilog implementations. See Appendix D for the complete testbench generation algorithm.

By combining these data preprocessing techniques—reasoning distillation, adaptive filtration, and testbench generation—with GRPO-based reinforcement learning, VeriReason addresses the core challenges of RTL generation: data scarcity, specification-code misalignment, and lack of self-verification. The framework enables models to internalize hardware design constraints through multi-level rewards and develop robust reasoning capabilities for RTL synthesis.

## 4 EXPERIMENTS

### 4.1 EXPERIMENTAL SETUP

Our primary dataset is derived from the RTLCoder Liu et al. (2024b) dataset of 26500 samples. We apply a thorough filtration technique on the dataset. First, we apply a simple syntax check to ensure we keep only the syntactically valid code. Then we use ChatGPT-4.1 to check whether the code matches the input prompt correctly, and generate reasoning steps for the code. For code that does not fully match the input prompt, it is re-generated and checked for syntax. We then generate a testbench for each entry with at least 100 test cases for best coverage.

The testbench output is also saved to the dataset; in this way, during the GRPO, the testbench will only run once on the generated code, and the outcome will be directly compared to the golden code's output. Next, we run the dataset on the Qwen2.5B model to generate the code and its corresponding rewards. Based on the reward, we split the dataset into simple and hard portions for the next training steps. We end up with 1149 samples in the hard level and 743 samples in the easy level.

Our evaluation focuses on the primary benchmarks for Verilog code generation, VerilogEval Liu et al. (2023b). The comprehensive benchmark containing both machine-generated and human-crafted Verilog specifications. VerilogEval-Machine contains 143 samples with algorithmically generated specifications, while VerilogEval-Human includes 156 samples with human-written specifications.

We evaluate VeriReason across multiple model scales to assess parameter efficiency, including Qwen2.5-1.5B, Qwen2.5-3B, Qwen2.5-7B, and CodeLlama3-7B architectures. GRPO is used as our default RL algorithm. RL-specific settings include a generation temperature of 0.5, a total batch size of 16 (8 rollouts each), an update batch size of 2 per GRPO step with a gradient accumulation of 8, a lowered learning rate of 1.0e-6 with constant scheduler type, and repetition penalty of 1.3. The reward model follows the design in Equation 2, with execution correctness verified using industry-standard Verilog simulators, Iverilog Williams (2023).

### 4.2 MAIN RESULTS

Table 1 compares `VeriReason` against state-of-the-art Verilog generation models. Our approach achieves superior performance across all model sizes, with VeriReason-Qwen2.5-7B demonstrating remarkable gains over base models: +17.1 and +24.0 percentage points on pass@1 for VerilogEval-Machine and VerilogEval-Human, respectively.

Even our smallest model, VeriReason-Qwen2.5-1.5B, outperforms many larger models after applying our reinforcement learning framework. The performance gains are particularly notable for smaller models, demonstrating the effectiveness of our approach in optimizing parameter efficiency. We

Table 1: Comparative analysis of Verilog code generation performance. Gray highlighting denotes the overall state-of-the-art results. (**bold**) indicates the best results in the model size category. Color-coded numbers show performance deltas relative to base models (green: improvement).

| Category | Method | Params. | Open Source | VerilogEval-Machine | | VerilogEval-Human | |
|---|---|---|---|---|---|---|---|
| | | | | pass@1 | pass@5 | pass@1 | pass@5 |
| **Base Model** | GPT-3.5-Turbo | N/A | ✗ | 63.5 | 78.0 | 31.2 | 47.0 |
| | GPT-4o-mini | N/A | ✗ | 66.0 | 72.4 | 54.2 | 62.0 |
| | GPT-4-Turbo | N/A | ✗ | 72.5 | 83.0 | 64.3 | **76.1** |
| | Qwen-2.5-Coder | 1.5B | ✓ | 25.6 | 40.8 | 8.3 | 17.9 |
| | Qwen2.5-Coder | 3B | ✓ | 48.4 | 58.9 | 21.3 | 32.7 |
| | Qwen2.5-Coder | 7B | ✓ | 52.7 | 69.7 | 23.9 | 41.1 |
| | CodeLlama | 7B | ✓ | 26.1 | 49.1 | 18.8 | 28.6 |
| | CodeQwen1.5-7B-Chat | 7B | ✓ | 29.1 | 61.9 | 14.8 | 36.8 |
| | DeepSeek-Coder | 6.7B | ✓ | 8.8 | 34.3 | 4.9 | 19.3 |
| | DeepSeek-V3 | 671B | ✓ | **79.2** | 80.7 | **66.1** | 72.1 |
| **Fine-tuned Generation** | ChipNeMo[†] | 70B | ✗ | 53.8 | N/A | 27.6 | N/A |
| | BetterV-CodeQwen[†] | 7B | ✗ | 68.1 | 79.4 | 46.1 | 53.7 |
| | RTLLLM[†] | 13B | ✗ | 65.3 | 77.2 | 43.7 | 51.8 |
| | VerilogEval[†] | 16B | ✗ | 46.2 | 67.3 | 28.8 | 45.9 |
| | VeriGen[†] | 16B | ✓ | 44.0 | 52.6 | 30.3 | 43.9 |
| | RTLCoder-DeepSeek-Coder | 6.7B | ✓ | $37.2_{+28.4}$ | $64.9_{+30.6}$ | $16.9_{+12.0}$ | $35.7_{+16.4}$ |
| **VeriReason (Ours)** | VeriReason-Qwen2.5-1.5B | 1.5B | ✓ | $\mathbf{44.7}_{+19.1}$ | $\mathbf{49.1}_{+8.3}$ | $\mathbf{23.5}_{+15.2}$ | $\mathbf{26.7}_{+8.8}$ |
| | VeriReason-Qwen2.5-3B | 3B | ✓ | $\mathbf{55.9}_{+7.5}$ | $\mathbf{72.8}_{+13.9}$ | $\mathbf{33.2}_{+11.9}$ | $\mathbf{47.4}_{+14.7}$ |
| | VeriReason-Qwen2.5-7B | 7B | ✓ | $\mathbf{69.8}_{+17.1}$ | $\mathbf{83.1}_{+13.4}$ | $\mathbf{47.9}_{+24.0}$ | $\mathbf{58.4}_{+17.3}$ |
| | VeriReason-codeLlama-7B | 7B | ✓ | $51.3_{+25.2}$ | $64.0_{+14.9}$ | $27.5_{+8.7}$ | $39.9_{+11.3}$ |

[†]: Reported Results.

observe that VeriReason-Qwen2.5-7B achieves state-of-the-art performance on Machine pass@5 (83.1%), outperforming even GPT-4-Turbo on this metric, despite having significantly less parameters.

Furthermore, all VeriReason models establish themselves as the **best performers** in their respective parameter size categories (1.5B, 3B, and 7B), highlighting the robustness of our approach across model scales. The substantial improvements observed in VeriReason-codeLlama-7B (+25.2 percentage points in Machine pass@1) further demonstrates that our method generalizes effectively across different model architectures.

### 4.3 Training Dynamics Analysis

Figure 2 illustrates the training dynamics of our VeriReason models across three different parameter scales (1.5B, 3B, and 7B). We track both the mean reward values (top row) and their standard deviations (bottom row) throughout the reinforcement learning process.

#### 4.3.1 Reward Progression

The reward curves reveal distinct learning patterns across model sizes. The 1.5B model demonstrates steady, monotonic improvement in reward values from approximately 0.5 to 0.8 over 800 training steps, suggesting a consistent optimization path. In contrast, the 3B model exhibits higher variance in its learning trajectory, with reward values fluctuating between 0.6 and 0.8, before ultimately converging to above 0.8 by step 400. The 7B model shows the most pronounced oscillatory behavior, with rewards ranging between 0.50 and 0.65, reflecting the increased complexity of optimizing larger parameter spaces.

#### 4.3.2 Reward Stability

The standard deviation plots (bottom row) provide critical insights into training stability. The 1.5B model demonstrates exceptional convergence properties, with reward variability consistently decreasing from 0.3 to below 0.1 throughout training. This smooth reduction in standard deviation correlates with the steady increase in mean rewards, indicating robust learning. The 3B model presents a more complex pattern, with initial rapid variability reduction followed by fluctuations between 0.7 and 0.8, suggesting periodic exploration-exploitation transitions. The 7B model's standard deviation

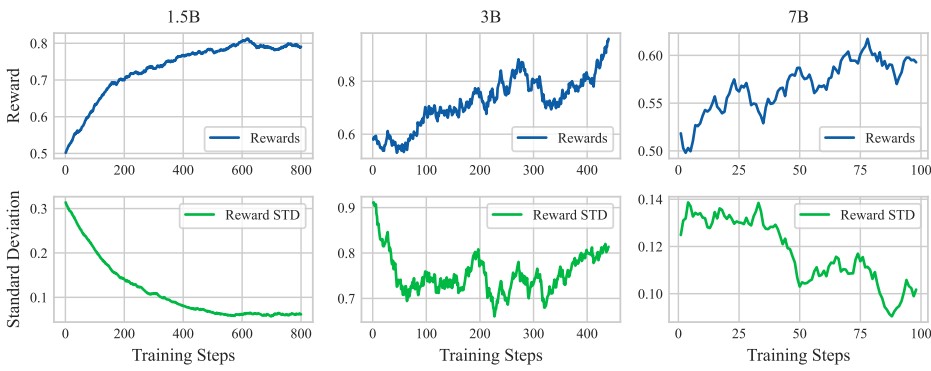

Figure 2: Reward line and std line of `VeriReason`

exhibits the most dynamic behavior, oscillating between 0.09 and 0.14, which aligns with its more variable reward progression.

Interestingly, despite having fewer parameters, the 1.5B model achieves the most stable convergence pattern, with monotonically decreasing standard deviation. This suggests that smaller models may benefit more consistently from our reinforcement learning framework, while larger models engage in more extensive exploration of the parameter space before convergence. The final standard deviation values (approximately 0.05 for 1.5B, 0.8 for 3B, and 0.1 for 7B) indicate that all models eventually reach stable policy configurations, though with different convergence trajectories.

This analysis provides empirical evidence that our reinforcement learning approach effectively optimizes models across different parameter scales, with larger models requiring more complex optimization paths but ultimately achieving higher reward values, consistent with their superior performance on the VerilogEval benchmarks shown in Table 1.

### 4.3.3 THE EFFECT OF SFT AND GRPO

Table 2 shows how SFT and GRPO contribute to VeriReason's performance. SFT provides substantial initial gains across all models, with smaller models benefiting most, by supplying high-quality training examples with reasoning steps. GRPO further improves performance through testbench-driven reinforcement learning. While SFT establishes syntactic and semantic understanding, GRPO enables functional correctness by internalizing structural constraints and self-checking capabilities, demonstrating their complementary roles.

Table 2: Ablation studies results.

| Model | Training Stage | VerilogEval-Machine | | VerilogEval-Human | |
|---|---|---|---|---|---|
| | | pass@1 | pass@5 | pass@1 | pass@5 |
| Qwen2.5-1.5B | Base | 25.6 | 40.8 | 8.2 | 17.9 |
| | + SFT | 38.6 | 46.3 | 17.8 | 23.9 |
| | + GRPO | **44.7** | **49.1** | **23.5** | **26.7** |
| Qwen2.5-3B | Base | 48.4 | 58.9 | 21.3 | 32.7 |
| | + SFT | 51.9 | 69.9 | 31.3 | 45.1 |
| | + GRPO | **55.9** | **72.8** | **33.2** | **47.4** |
| Qwen2.5-7B | Base | 52.7 | 69.7 | 23.9 | 41.1 |
| | + SFT | 63.4 | 79.9 | 43.4 | 56.2 |
| | + GRPO | **69.8** | **83.1** | **47.9** | **58.4** |
| CodeLlama-7B | Base | 26.1 | 49.1 | 18.8 | 28.6 |
| | + SFT | 41.1 | 58.3 | 23.2 | 31.5 |
| | + GRPO | **51.3** | **64.0** | **27.5** | **39.9** |

## 5 CONCLUSION

This paper presents `VeriReason`, a framework that integrates supervised fine-tuning with GRPO-based reinforcement learning for Verilog RTL generation. By combining explicit reasoning with testbench-driven feedback, we address key challenges in LLM-based hardware design: data scarcity, specification-code misalignment, lack of self-checking, and insufficient logical reasoning. `VeriReason` achieves state-of-the-art performance across model scales, though with computational overhead during training and 2.5-3× slower inference. Our multi-level reward model—combining functional, structural, and syntactic validation—enables models to develop intrinsic self-checking capabilities crucial for autonomous hardware design. The consistent improvements across different architectures demonstrate the robustness of our approach. By open-sourcing our models and datasets, we aim to accelerate progress in automated RTL generation and contribute to the community.

# 6 ETHICS STATEMENT

The primary goal of this research is to advance automated hardware design through improved RTL code generation techniques, potentially reducing development time and human error in digital circuit design. Our work introduces `VeriReason`, a framework that combines reinforcement learning with reasoning capabilities to generate functionally correct Verilog code. We emphasize that our research is conducted purely for academic and technological advancement, with no intent to replace human hardware engineers but rather to augment their capabilities and productivity.

The automated generation of hardware descriptions raises no immediate ethical concerns beyond standard considerations for AI-assisted design tools. Our framework generates code that must still undergo standard verification and testing procedures before deployment in any real-world system. We do not target safety-critical systems or applications where hardware failures could endanger human life without appropriate human oversight and validation.

All datasets used in this work are derived from publicly available sources or generated synthetically, with no proprietary or confidential design information involved. We believe this research contributes positively to the hardware design community by democratizing access to advanced RTL generation capabilities and reducing the barrier to entry for hardware development.

# 7 REPRODUCIBILITY STATEMENT

To ensure full reproducibility of our research, we release our code, datasets, and trained models upon acceptance. The implementation details necessary for reproduction include:

**Dataset:** The methodology for creating our training dataset from the RTLCoder dataset is detailed in Section 3.3. We provide the complete data filtering pipeline, reasoning generation prompts, and testbench generation scripts. Our filtered dataset consists of 1,892 high-quality samples (1,149 hard, 743 simple) with reasoning steps and testbenches.

**Training Configuration:** All hyperparameters are specified in Section 4.1, including: learning rate (1.0e-6), batch size (16), temperature (0.5), GRPO rollouts (8), and reward weights (functional: 2.0, AST: 1.0, syntax: 0.1). Model architectures tested include Qwen2.5 (1.5B, 3B, 7B) and CodeLlama-7B.

**Evaluation:** We use the standard VerilogEval benchmark (Machine: 143 samples, Human: 156 samples) with pass@1 and pass@5 metrics. Evaluation scripts using Iverilog for syntax checking and functional verification will be provided.

All experiments were conducted on NVIDIA RTX 6000 Ada GPUs. The complete codebase, including data preprocessing, model training, and evaluation scripts, is available at https://anonymous.4open.science/r/VeriReason-E625. We estimate the total computational cost for reproducing our main results at approximately 200 GPU hours.

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

## A   LLM USAGE STATEMENT

In the development of this work, we employed several LLMs at different stages to enhance research efficiency and quality. The specific applications are as follows:

**Dataset Curation and Quality Control.** After establishing the initial VeriReason framework, we used GPT-4 to verify the correctness of code-specification alignments in our dataset and to generate initial reasoning steps for each sample. These outputs were thoroughly reviewed and refined by the authors to ensure accuracy. The LLM-generated reasoning served as a starting point that was iteratively improved through our reasoning-distillation pipeline described in Section 3.3.2.

**Experimental Analysis.** During the evaluation phase, we used LLMs to help identify potential failure modes in generated Verilog code and to suggest additional test cases for our testbench generation pipeline. This process helped improve the robustness of our evaluation methodology, though all final decisions and implementations were made by the authors.

**Manuscript Polishing.** We used LLMs as writing assistants to improve sentence structure, check technical terminology consistency, and enhance manuscript clarity. All LLM-generated content was critically reviewed and verified by the authors.

In line with ICLR 2026 policy on LLM usage, we note that all analyses, design choices, and interpretations of results was thoroughly reviewed, validated, and often substantially modified by the authors, who take full responsibility for the scientific integrity and accuracy of this paper. LLMs served purely as tools to enhance productivity and are not authors of this work.

## B   GRPO ADVANTAGE ESTIMATION FOR MULTI-LEVEL REWARDS

For group size $G$ with rewards distributed across syntax errors $(n_0)$, partial correctness $(n_p)$, and full correctness $(n_f)$, the variance comparison shows:

$$\text{Var}_{\text{multi}} = \frac{1}{G}\left[n_0(0-\mu)^2 + n_p\text{Var}[\text{AST}] + n_f(2-\mu)^2\right]$$

$$< \text{Var}_{\text{binary}} = \frac{1}{G}\left[(n_0+n_p)(0-\mu_b)^2 + n_f(2-\mu_b)^2\right]$$

where $\mu_b = 2n_f/G$ for binary rewards and $\mu$ includes AST contributions. Since $\text{Var}[\text{AST}] < (0-\mu_b)^2$, multi-level rewards reduce gradient variance.

Gradient preservation: Binary rewards give $\nabla_\theta J = 0$ for syntactically correct but functionally incorrect samples, while multi-level rewards provide $\nabla_\theta J \propto \text{ASTscore}(o) - \bar{\text{AST}}$, ensuring continuous learning signals.

Convergence rate: $\|\theta_{k+1} - \theta^*\| \leq \rho^k \|\theta_0 - \theta^*\|$ where $\rho_{\text{multi}} < \rho_{\text{binary}}$ due to continuous AST guidance.

## C   AST-BASED STRUCTURAL SIMILARITY CALCULATION

### C.1   AST EXTRACTION ALGORITHM

The AST-based structural similarity score is computed by extracting and comparing key structural elements from both generated and golden Verilog code. The extraction process identifies the following components:

**Algorithm 1** Extract AST Statements from Verilog Code

1: **Input:** Verilog code string $code$
2: **Output:** List of AST statements $statements$
3: Remove all comments from $code$
4: Initialize $statements \leftarrow []$
5: **Extract Module Declaration:**
6:  Find module name and port list
7:  $statements$.append("module {name}")
8:  For each port: $statements$.append("port {port}")
9: **Extract I/O Declarations:**
10:  For each $type \in$ {input, output, inout}:
11:   Find all declarations of $type$
12:   $statements$.append("$type$ [width] {name}")
13: **Extract Variable Declarations:**
14:  For each $type \in$ {wire, reg}:
15:   Extract width and name
16:   $statements$.append("$type$ [width] {name}")
17: **Extract Assign Statements:**
18:  For each assign statement:
19:   $statements$.append("assign {lhs} = {rhs}")
20: **Extract Always Blocks:**
21:  For each always block:
22:   Extract sensitivity list
23:   $statements$.append("always@({sensitivity})")
24:   Extract control flow (if/else/case) within block
25: **Extract Module Instantiations:**
26:  For each instance:
27:   $statements$.append("instance {type} {name}")
28: **Return** $statements$

## C.2 SIMILARITY SCORE CALCULATION

Given the extracted statements from generated code ($S_g$) and golden code ($S_{gold}$), the final AST score is computed as:

**Algorithm 2** Compute AST Similarity Score

1: **Input:** Generated statements $S_g$, Golden statements $S_{gold}$
2: **Output:** AST similarity score $score \in [0, 1]$
3: Compute sequence similarity: $sim \leftarrow$ SequenceMatcher($S_g$, $S_{gold}$).ratio()
4: Convert to sets: $Set_g \leftarrow$ set($S_g$), $Set_{gold} \leftarrow$ set($S_{gold}$)
5: Compute coverage: $cov \leftarrow \frac{|Set_g \cap Set_{gold}|}{|Set_{gold}|}$
6: Compute redundancy: $red \leftarrow \frac{|Set_g - Set_{gold}|}{|Set_g|}$
7: Calculate weighted score: $score \leftarrow 0.7 \cdot sim + 0.3 \cdot cov - 0.2 \cdot red$
8: Clamp to valid range: $score \leftarrow \max(0, \min(1, score))$
9: **Return** $score$

## C.3 SCORE COMPUTATION

Given generated statements set $S_g$ and golden statements set $S_{gold}$:

$$sim = \text{LevenshteinRatio}(S_g, S_{gold}) \tag{4}$$

$$cov = \frac{|S_g \cap S_{gold}|}{|S_{gold}|} \tag{5}$$

$$red = \frac{|S_g \setminus S_{gold}|}{|S_g|} \tag{6}$$

$$\text{ASTscore} = \min\left(1, \max\left(0, 0.7 \cdot sim + 0.3 \cdot cov - 0.2 \cdot red\right)\right) \tag{7}$$

where $S_g \setminus S_{gold}$ denotes set difference (elements in $S_g$ but not in $S_{gold}$).

## D  AUTOMATED TESTBENCH GENERATION

### D.1  TESTBENCH GENERATION PIPELINE

Our automated testbench generation creates comprehensive test suites for Verilog modules by analyzing specifications and generating diverse test vectors.

---

**Algorithm 3** Generate Testbench for Verilog Module

---

1: **Input:** Module specification $S$, Module interface $I$
2: **Output:** Testbench code $TB$, Expected outputs $E$
3: **Parse Module Interface:**
4:     Extract input ports: $inputs \leftarrow \text{ParseInputs}(I)$
5:     Extract output ports: $outputs \leftarrow \text{ParseOutputs}(I)$
6:     Identify signal widths and types
7: **Identify Test Requirements:**
8:     Extract functional requirements from $S$
9:     Determine boundary conditions
10:     Identify edge cases from specification
11: **Generate Test Vectors:**
12:     $vectors \leftarrow []$
13:     **// Directed Testing**
14:     For each explicit requirement in $S$:
15:         Generate specific test cases
16:         $vectors$.append(directed_tests)
17:     **// Boundary Testing**
18:     For each input signal:
19:         Add min/max values: $\{0, 2^{width} - 1\}$
20:         Add boundary transitions
21:     **// Constrained Random Testing**
22:     While $|vectors| < 100$:
23:         Generate random inputs respecting constraints
24:         Ensure coverage of untested combinations
25:         $vectors$.append(random_test)
26: **Generate Testbench Structure:**
27:     Create module instantiation
28:     Initialize test vector application loop
29:     Add output monitoring and logging
30:     Write results to "test_vectors.txt"
31: **Validate Testbench:**
32:     Run against golden implementation
33:     Verify coverage $\geq$ threshold
34:     Confirm detection of known-bad implementations
35: **Return** $TB$, $E$

---

## D.2 TEST VECTOR GENERATION STRATEGIES

The testbench employs three complementary strategies to ensure comprehensive coverage:

---

**Algorithm 4** Multi-Strategy Test Vector Generation

---

1: **function** GenerateTestVectors($inputs$, $spec$, $min\_count = 100$)
2:     $test\_vectors \leftarrow []$
3:     **// Strategy 1: Directed Tests (30% of vectors)**
4:     For each functional requirement $req$ in $spec$:
5:       $test\_vectors$.append(CreateDirectedTest($req$))
6:     **// Strategy 2: Edge Cases (20% of vectors)**
7:     For each input $in$ in $inputs$:
8:       $test\_vectors$.append($\{in \mapsto 0, \text{others} \mapsto \text{random}\}$)
9:       $test\_vectors$.append($\{in \mapsto 2^{width(in)} - 1\}$)
10:       If $in$ is clock/control signal:
11:         Add transition tests
12:     **// Strategy 3: Random Coverage (50% of vectors)**
13:     $coverage\_map \leftarrow$ InitializeCoverageGrid($inputs$)
14:     While $|test\_vectors| < min\_count$:
15:       $vector \leftarrow$ GenerateConstrainedRandom($inputs$, $spec$)
16:       If $vector$ improves $coverage\_map$:
17:         $test\_vectors$.append($vector$)
18:         UpdateCoverage($coverage\_map$, $vector$)
19:     **Return** $test\_vectors$

---

## D.3 COVERAGE METRICS

Test vector coverage is validated using:

$$\text{Functional Coverage} = \frac{|\text{Tested Requirements}|}{|\text{Total Requirements}|} \tag{8}$$

$$\text{Input Coverage} = \frac{|\text{Unique Input Combinations}|}{|\text{Possible Combinations}|} \tag{9}$$

$$\text{Boundary Coverage} = \frac{|\text{Tested Boundaries}|}{2 \cdot |\text{Input Signals}|} \tag{10}$$

## E  SAMPLE TRAINING DATASET

The following section demonstrates a sample entry from the training dataset used to train the hardware optimization models. Each dataset entry contains a problem specification, implementation requirements, and comprehensive test vectors for validation.

---

**Problem Specification**

You are tasked with creating a module that takes in two 16-bit unsigned numbers, a and b, and outputs their sum. The sum output should be a 17-bit unsigned number to prevent overflow. You are not allowed to use any built-in arithmetic operators or functions.

Create a module named `adder` that has the following inputs and outputs:

- **Input** a: a 16-bit unsigned number, **Input** b: a 16-bit unsigned number
- **Output** sum: a 17-bit unsigned number representing the sum of a and b

You may use any other verilog constructs to solve this problem.

---

Figure 3: Training Dataset Entry: Problem Specification

---

**1. Requirement Analysis:**
The module must compute the sum of two 16-bit unsigned inputs (a, b) and output a 17-bit unsigned sum. No built-in adders/operators may be used.
**2. Logical Design Decisions:**
Manual bit-wise addition is necessary. For each bit, XOR forms the sum (with the carry-in), while carry-out is calculated using bit-wise AND and carried forward. This ensures proper binary addition and propagation of carries without +. The final carry becomes the most significant bit.
**3. Implementation Choices:**
Combinational logic is appropriate because the output should update immediately as inputs change. Direct assignment with 'assign' since this is best for power gating and quick changes.

This approach respects all constraints (no arithmetic ops), handles all bits and carry correctly, and produces the required output in a clear hardware manner.

---

Figure 4: Training Dataset Entry: Design Reasoning

---

```verilog
module adder(
input [15:0] a,
input [15:0] b,
output reg [16:0] sum
);

integer i;
reg carry;
always @(*) begin
carry = 1'b0;
// Loop through each bit
for (i = 0; i < 16; i = i + 1) begin
sum[i] = a[i] ^ b[i] ^ carry;
carry = (a[i] & b[i]) | (a[i] & carry) | (b[i] & carry);
end
sum[16] = carry;
end

endmodule
```

*Listing 1:* Training Dataset Entry: Reference Implementation.

```verilog
1  module adder_tb;
2
3  // Inputs
4  reg [15:0] a;
5  reg [15:0] b;
6
7  // Outputs
8  wire [16:0] sum;
9
10 // DUT instantiation
11 adder uut (
12 .a(a),
13 .b(b),
14 .sum(sum)
15 );
16
17 // File descriptor
18 integer fd;
19 integer i;
20 integer error_count;
21 reg [16:0] expected_sum;
22
23 // Task to check result and report errors
24 task check_result;
25 input [15:0] a_in;
26 input [15:0] b_in;
27 input [16:0] sum_out;
28 input [16:0] expected;
29 begin
30 if (sum_out !== expected) begin
31 $display("ERROR: a=%h b=%h sum=%h expected=%h at time %0t",
32          a_in, b_in, sum_out, expected, $time);
33 error_count = error_count + 1;
34 end
35 end
36 endtask
37
38 // Main stimulus with corner cases and randomized tests
39 initial begin
40 error_count = 0;
41 fd = $fopen("test_vectors.txt", "w");
42
43 // Corner case tests
44 a = 16'h0000; b = 16'h0000; #1; // All zeros
45 expected_sum = a + b;
46 $fdisplay(fd, "%h %h %h", a, b, sum);
47 check_result(a, b, sum, expected_sum);
48
49 a = 16'hFFFF; b = 16'hFFFF; #1; // Maximum overflow
50 expected_sum = a + b;
51 $fdisplay(fd, "%h %h %h", a, b, sum);
52 check_result(a, b, sum, expected_sum);
53
54 // Randomized comprehensive testing
55 for (i = 0; i < 90; i = i + 1) begin
56 a = $random;
57 b = $random;
58 #1;
59 expected_sum = a + b;
60 $fdisplay(fd, "%h %h %h", a, b, sum);
61 check_result(a, b, sum, expected_sum);
62 end
63
64 $display("Testing complete. Total errors: %0d", error_count);
65 $fclose(fd);
66 $finish;
67 end
68
69 endmodule
```

*Listing 2:* Training Dataset Entry: Comprehensive Testbench.

```
1  A     B     Sum
2 ---------------
3 0000  0000  00000
4 ffff  ffff  1fffe
5 0000  ffff  0ffff
6 ffff  0000  0ffff
7 8000  8000  10000
8 7fff  0001  08000
9 1234  4321  05555
10 aaaa  5555  0ffff
11 0001  0001  00002
12 8000  7fff  0ffff
13 3524  5e81  093a5
14 d609  5663  12c6c
15 7b0d  998d  1149a
16 8465  5212  0d677
17 e301  cd0d  1b00e
18 f176  cd3d  1beb3
19 57ed  f78c  14f79
20 e9f9  24c6  10ebf
21 84c5  d2aa  1576f
22 f7e5  7277  16a5c
23 d612  db8f  1b1a1
24 69f2  96ce  100c0
25 7ae8  4ec5  0c9ad
26 495c  28bd  07219
27 582d  2665  07e92
28 6263  870a  0e96d
29 2280  2120  043a0
30 45aa  cc9d  11247
31 3e96  b813  0f6a9
32 380d  d653  10e60
33 dd6b  2ad5  10840
34 4a02  3eae  088b0
35 e91d  72cf  15bec
36 4923  650a  0ae2d
37 0aca  4c3c  05706
38 bdf2  618a  11f7c
39 b341  34d8  0e819
40 f378  1289  10601
41 0deb  65b6  073a1
42 f9c6  13ae  10d74
43 02bc  dd2a  0dfe6
44 9a0b  be71  1587c
45 4185  554f  096d4
46 603b  333a  09375
47 327e  4b15  07d93
48 9bf1  4bd9  0e7ca
49 0762  fb4c  102ae
50 559f  a18f  0f72e
51 a9f8  60b7  10aaf
52 569f  945c  0eafb
53 c05b  3789  0f7e4
54 3249  3ed0  07119
55 c0d7  fc51  1bd28
56 2f96  7f0c  0aea2
57 cec2  edc8  1bc8a
58 5a77  ed3d  147b4
59 db12  007e  0db90
60 816d  e739  168a6
61 8f1f  f6d3  185f2
62 2f85  8878  0b7fd
63 595b  4b49  0a4a4
64 ae3f  af2a  15d69
65 6358  3886  09bde
66 0c8e  f29c  0ff2a
67 99fa  bc26  15620
68 1773  b4a3  0cc16
```

*Listing 3:* Training Dataset Entry: Generated Test Vectors (Format: Input_A Input_B Expected_Sum).