# OpenReview forum: "VeriReason: Reinforcement Learning with Testbench Feedback for Reasoning-Enhanced Verilog Generation"
_ICLR.cc/2026/Conference — Submitted to ICLR 2026_

### Official Review · Reviewer_4H7J · 2025-10-30

**Soundness:** 3
**Presentation:** 3
**Contribution:** 2
**Rating:** 4
**Confidence:** 4

**Summary:**

This paper introduces VeriReason, a framework for automated Verilog RTL code generation that integrates supervised fine-tuning with Group Relative Policy Optimization (GRPO)-based reinforcement learning. The framework uniquely combines structural, functional, and syntactic reward signals for optimizing large language models in hardware design domains. VeriReason is empirically shown to achieve state-of-the-art results on the VerilogEval benchmark, notably outperforming both open and commercial baselines across multiple model sizes. The approach emphasizes alignment with specification, self-checking behavior, and improved generalization with significantly less training data.

**Strengths:**

1. The integration of explicit testbench-driven feedback within a reinforcement learning loop (GRPO) for Verilog code generation is carefully engineered and directly tailored to the domain;
2. The work includes comprehensive ablations that disentangle and validate the individual and combined effects of supervised fine-tuning and GRPO, making the contribution measurable and transparent;
3. This paper open-sources its codebase and curated dataset to advance reproducibility and benchmarking practices in the research community.

**Weaknesses:**

1. In Table 1, baselines such as CodeV[1] and CraftRTL[2] for RTL generation are missed, which demonstrate stronger performance on the VerilogEval benchmark;
2. There are other concurrent works on reinforcement learning-based RTL generation, such as [3] and [4]. It would be better if the authors can give a brief review of these works and clarify the difference between this work and others;
3. The paper includes several manually selected hyperparameters. For instance, on lines 299–300, could the authors elaborate on the intuition or rationale behind the chosen values for $\alpha_{\text{min}}$, $\alpha_{\text{max}}$ and $\beta$? Similarly, on line 272, how are the weights 0.6, 0.5, and -0.3 selected? Finally, how sensitive are the overall results to variations in these hyperparameters?

[1] Zhao, Y., Huang, D., Li, C., Jin, P., Song, M., Xu, Y., Nan, Z., Gao, M., Ma, T., Qi, L. and Pan, Y., 2025. Codev: Empowering llms with hdl generation through multi-level summarization. *IEEE Transactions on Computer-Aided Design of Integrated Circuits and Systems*.

[2] Liu, M., Tsai, Y.D., Zhou, W. and Ren, H., 2024. Craftrtl: High-quality synthetic data generation for verilog code models with correct-by-construction non-textual representations and targeted code repair. *arXiv preprint arXiv:2409.12993*.

[3] Teng, F., Pan, M., Zhang, X., He, Z., Yang, Y., Chai, X., Qi, M., Lu, L. and Yin, J., 2025. VERIRL: Boosting the LLM-based Verilog Code Generation via Reinforcement Learning. *arXiv preprint arXiv:2508.18462*.

[4] Wang, N., Yao, B., Zhou, J., Hu, Y., Wang, X., Guan, N. and Jiang, Z., 2025. Insights from verification: Training a verilog generation LLM with reinforcement learning with testbench feedback. *arXiv preprint arXiv:2504.15804*.

**Questions:**

In Table 1, the reported performance for RTLCoder-DeepSeek-Coder on VerilogEval-Machine is 37.2 for pass@1 and 64.9 for pass@5, while on VerilogEval-Human it is 16.9 for pass@1 and 35.7 for pass@5. These results exhibit a substantial discrepancy compared to the original RTLCoder paper, which reports 61.2 for pass@1 and 76.5 for pass@5 on VerilogEval-Machine, and 41.6 for pass@1 and 50.1 for pass@5 on VerilogEval-Human. Could the authors elaborate on the reasons for this gap?

---

### Official Review · Reviewer_gnu5 · 2025-10-31

**Soundness:** 3
**Presentation:** 3
**Contribution:** 2
**Rating:** 4
**Confidence:** 4

**Summary:**

This paper introduces a new framework for Verilog Code Generation. The proposed VeriReason integrates Supervised Fine-Tuning (SFT) and Group Relative Policy Optimization (GRPO) reinforcement learning to enhance the performance of Verilog Completion. The key contributions of this paper include
(1)	Collect a high-quality Verilog dataset of 1,892 samples with a reasoning trajectory.
(2)	Design the reward method with testbench feedback using syntactic correctness, functional correctness, and structural similarity.

**Strengths:**

(1)	A filtering algorithm for Verilog corpora. A major contribution of this paper is the two-stage adaptive filtration process to collect Verilog modules. These complex steps ensure the stability of GRPO training, especially the combination of the reward function.

(2)	A reward model with reinforcement learning testbench feedback. The reward score includes three measures from syntactic correctness, functional correctness, and structural similarity. Selected hyperparameters ensure the balance of these three components. During GRPO training, this scoring system prevents gradient vanishing and provides faster convergence.

(3)	Reasoning generation and testbench generation. ChatGPT-4.1 is used to generate reasoning steps. The distillation trajectory could improve the reasoning ability of SFT model. Algorithm 3 is effective for automated testbench generation.

**Weaknesses:**

(1) Limited novelty and insights. Regarding the methodology, the authors are only performing SFT and GRPO on Qwen2.5, without proposing new training paradigms specifically for Verilog code generation or addressing some unique challenges. The results (in Fig. 2) only showed increasing reward scores, but an increasing reward may be achieved by the model's reward hacking, instead of true improvement in coding ability. Better show the real scores on benchmarks at different training steps. In addition, it is not sufficiently clear which specific policy contributed to the higher performance, compared with the SFT baseline.

(2) Insufficient experiments for both benchmark and baselines. The authors only use the VerilogEval dataset to evaluate the pass@k. For Verilog Code Generation, the common benchmarks in most papers also include RTLLM [1] and CVDP [2]. More importantly, this paper has missed several recent works with 7B models that achieve higher scores on VerilogEval-Machine and VerilogEval-Human, such as CodeV [3] and CraftRTL [4]. Also, it seems to be a serious mistake that RTLLM [1], as a widely adopted benchmark, is listed as a model (as RTLLLM) with performance values in Table 1. In addition, regarding commericial LLMs, models such as GPT-5 can also be included.

(3) Although the filtering and forming of the Verilog dataset is useful to the community, there is very limited quantitative analysis on these 1,892 samples. In addition, there have been many existing datasets (with reasoning) for Verilog code generation, and some may be of higher quality or greater quantity.

(4) Possibly missing content. Seems the reviewer cannot find results corresponding to these claimed contributions: "2.8× increase in first-attempt functional correctness", “even with as few as 20 annotated examples, GRPO yields substantial performance gains”.

(5) It may be worthwhile to change the base model from Qwen2.5 to other models. The solution may differ greatly when evaluated on different base models.

[1] RTLLM: An open-source benchmark for design RTL generation with large language model.
[2] Comprehensive Verilog Design Problems: A Next-Generation Benchmark Dataset for Evaluating Large Language Models and Agents on RTL Design and Verification.
[3] CodeV: Empowering LLMs with HDL Generation through Multi-Level Summarization.
[4] CraftRTL: High-quality synthetic data generation for Verilog code models with correct-by-construction non-textual representations and targeted code repair.

**Questions:**

Please see the weakness part above.

---

### Official Review · Reviewer_ZrAm · 2025-11-01

**Soundness:** 2
**Presentation:** 2
**Contribution:** 2
**Rating:** 2
**Confidence:** 4

**Summary:**

The paper introduces VeriReason, which combines SFT with GRPO to generate Verilog RTL code. The approach uses testbench feedback and AST-based structural rewards to train models that can reason through hardware design problems. The method is evaluated on VerilogEval benchmarks using various model sizes from 1.5B to 7B parameters.

Though the paper represents an admirable contribution to this growing field, it presents several problems 1) the dataset and reproducibility details are incomplete, 2) the paper fails to clearly justify why GRPO is better than other RL approaches: for example, while the authors claim GRPO is more efficient than PPO, there's no empirical comparison between the two methods, 3) the experiments are missing crucial details about benchmark versions (which VerilogEval?) and baseline configs.

The authors could improve the chances of their paper acceptance by clarify the exact VerilogEval version and provide results on other popular benchmarks (CVDP, RTLLLM, etc) and add comparisons with other RL methods. Willing to reconsider the scores if these questions are addressed.

**Strengths:**

- The authors applies GRPO to a novel problem in data scarce hw frontend design with a well-designed multi-level reward system
- The paper showcases impressive results: 83.1% pass@5 on VerilogEval-Machine, outperforming GPT-4 Turbo (83.0%) with much smaller models. The improvements are particularly impressive for smaller models (1.5B: +19.1 points).
-- However, this is also a weakness, as VerilogEval-Human numbers lag significantly behind.
- The adaptive data filtration strategy (retaining samples with mean reward ∈ [0.3, 1.8] and std > 0.1) is clever for identifying learnable examples.

**Weaknesses:**

- The paper mentions using VerilogEval but doesn't specify which version (v1 or v2); also failed to explain the differences in model performance for VerilogEval-Machine and VerilogEval-Human. These are important because impressive achievement in the former could be results of eval data contamination, since it was scraped from problems online
- Where are the evaluation results for RTLLM and similar benchmarks?
- No ablation on GRPO vs other RL algorithms (PPO, DPO)
- Using GPT4 to regenerate and check code could introduce biases from GPT 4's training data, reinforcing the eval data contamination issue
- The training curves (Figure 2) show very different x-axis scales 800 vs 400 vs 100 steps, varied with model sizes, without explanation

**Questions:**

- Which VerilogEval version are you using?

- Why the specific reward values (0.1, 1.1, 2.0)? The jump from max AST score (1.1) to functional correctness (2.0) seems arbitrary

- Where are the RL baselines? Table 1 only compares against SFT methods. Where's the comparison with other RL approaches on the same task? Why is GRPO necessary?

- The paper says that testbenches are generated automatically. How do you validate they actually test the functional correctness of the RTL code? What's the coverage of these generated tests?

---

### Meta-Review · Area_Chair_CHjW · 2025-12-20

**Summary:**

This paper introduces VeriReason, a new framework for SFT and RL with GRPO for RTL (Register Transfer Level) code generation. The authors collected high-quality training examples with feedback-based reward model. Specifically, VeriReason introduces testbench evaluations with heuristics to improve code alignment and reduce hallucination. The authors evaluated on the VerilogEval Benchmark and observed improvement in performance against strong models like GPT4 Turbo.

**Reviewer Concerns:**

1. There is a lacking in experimental details, including the version of VerilogEval (v1 or v2) and missing strong baselines such as CodeV and CraftRTL. The authors did not experiment on other major benchmarks for Verilog code generation tasks that a reviewer suggested.
2. Missing experimental results of GRPO against other major RL methods such as PPO and DPO. Moreoever, the results are mostly based on Qwen2.5 model series.
3. There is potential biases in using GPT4 to regenerate code and check the code from GPT4's training data
4. The paper has limited novelty as the proposed method simply combines SFT and RL with GRPO for Verilog code generation tasks.

**Reviewer Scores:**

The authors did not engage in the discussion during the rebuttal. Therefore, the reviewers would not change their scores.

---

### Decision · Program_Chairs · 2026-01-26

Reject